# Two-Month Voluntary Ethanol Consumption Promotes Mild Neuroinflammation in the Cerebellum but Not in the Prefrontal Cortex, Hippocampus, or Striatum of Mice

**DOI:** 10.3390/ijms25084173

**Published:** 2024-04-10

**Authors:** Pablo Berríos-Cárcamo, Sarah Núñez, Justine Castañeda, Javiera Gallardo, María Rosa Bono, Fernando Ezquer

**Affiliations:** 1Center for Regenerative Medicine, Faculty of Medicine, Clínica Alemana-Universidad del Desarrollo, Santiago 7610615, Chile; javiera.gallardo@udd.cl (J.G.); eezquer@udd.cl (F.E.); 2Facultad de Medicina y Ciencia, Universidad San Sebastián, Sede Los Leones 7510602, Chile; sarah.nunez@uss.cl; 3Centro Ciencia & Vida, Santiago 8580702, Chile; 4Departamento de Biología, Facultad de Ciencias, Universidad de Chile, Santiago 7800003, Chile; justine.castaneda@usach.cl (J.C.); mrbono@uchile.cl (M.R.B.); 5Research Center for the Development of Novel Therapeutics Alternatives for Alcohol Use Disorders, Santiago 7610658, Chile

**Keywords:** ethanol, neuroinflammation, MCP1, cerebellum, microglia

## Abstract

Chronic ethanol exposure often triggers neuroinflammation in the brain’s reward system, potentially promoting the drive for ethanol consumption. A main marker of neuroinflammation is the microglia-derived monocyte chemoattractant protein 1 (MCP1) in animal models of alcohol use disorder in which ethanol is forcefully given. However, there are conflicting findings on whether MCP1 is elevated when ethanol is taken voluntarily, which challenges its key role in promoting motivation for ethanol consumption. Here, we studied MCP1 mRNA levels in areas implicated in consumption motivation—specifically, the prefrontal cortex, hippocampus, and striatum—as well as in the cerebellum, a brain area highly sensitive to ethanol, of C57BL/6 mice subjected to intermittent and voluntary ethanol consumption for two months. We found a significant increase in MCP1 mRNA levels in the cerebellum of mice that consumed ethanol compared to controls, whereas no significant changes were observed in the prefrontal cortex, hippocampus, or striatum or in microglia isolated from the hippocampus and striatum. To further characterize cerebellar neuroinflammation, we measured the expression changes in other proinflammatory markers and chemokines, revealing a significant increase in the proinflammatory microRNA miR-155. Notably, other classical proinflammatory markers, such as TNFα, IL6, and IL-1β, remained unaltered, suggesting mild neuroinflammation. These results suggest that the onset of neuroinflammation in motivation-related areas is not required for high voluntary consumption in C57BL/6 mice. In addition, cerebellar susceptibility to neuroinflammation may be a trigger to the cerebellar degeneration that occurs after chronic ethanol consumption in humans.

## 1. Introduction

Ethanol is one of the most consumed addictive drugs. Globally, 43% of the population aged 15 years and older are current ethanol drinkers [1]. Prolonged and excessive ethanol consumption can lead to compulsive drinking, characterized by a loss of control over the consumption of large amounts of ethanol despite negative consequences. This ailment is known as an alcohol use disorder (AUD), a chronic relapsing condition of ethanol dependency that affects approximately 5% of the global adult population [1,2] and has afflicted 29% of adults in the US at some point in their lives [3].

Recently, there has been a growing focus on investigating brain proinflammatory alterations in individuals with AUD, as evidence suggests that the ensuing neuroinflammation contributes to the compulsive consumption of ethanol [4,5,6,7,8]. Over the past 15 years, studies employing animal models of AUD have shown that an innate immune response occurs in the brain secondary to ethanol exposure, dependent on the activation of sensors of foreign molecules such as the Toll-like receptor 4 (TLR4) [9]. This interaction promotes the activation of microglia, the primary neuroimmune cells, which course morphological changes that are usually observed by the rise in the levels of the cytoskeleton-related protein ionized calcium-binding adaptor molecule 1 (Iba1), a promoter of actin crosslinking [10]. Microglia activation results in the nuclear translocation of NF-κB, promoting the expression of proinflammatory genes, resulting in the rise of proinflammatory cytokines at mRNA and protein levels and of proinflammatory microRNAs [11]. These phenomena are usually observed in animal models of AUD after ethanol exposure and in postmortem tissue samples of AUD patients, in brain areas related to ethanol consumption comprising the brain reward system (see below), specifically the mesocorticolimbic and nigrostriatal systems that control motivation [12,13] and the hippocampus that controls consumption learning and memory [14]. 

Among the proinflammatory markers, the monocyte chemoattractant protein 1 (MCP1) stands out as it has shown the most consistent results. MCP1 is a chemokine known for promoting microglia proinflammatory activation and their recruitment to affected areas in different neuroinflammatory diseases [15], a role that is also observed after ethanol exposure [16,17]. Likewise, increased levels of MCP1 or its mRNA have been reported in the brain of mouse [18,19,20,21,22,23,24,25], rat [26,27,28], and non-human primate [29] models of AUD. However, in all these studies, ethanol intake was forced, thus a direct role of MCP1 in the motivation for ethanol consumption cannot be directly assigned. 

While studies involving ethanol-exposed animals with ethanol provided via a liquid diet as the sole food source, as an ethanol solution as the sole liquid source, or directly administered by gavage, consistently demonstrate neuroinflammation [18,19,20,21,22,23,24,25,26,27,28,29,30,31,32,33], studies in which ethanol consumption is voluntary yield variable outcomes. For example, an RNAseq analysis of the nucleus accumbens in rats that voluntarily consumed ethanol for 8 weeks showed no significant differences compared to control animals [34]. Similarly, 20 days of voluntary ethanol consumption of scaling concentrations caused a reduction in microglia Iba1 immunoreactivity in the hippocampus of ethanol-exposed rats [35], instead of the expected increase. In mice, the RNAseq profile of prefrontal cortex homogenates or isolated microglia from mice that intermittently and voluntarily consumed ethanol showed significant variations in the expression of immune-related genes, though classical proinflammatory markers such as MCP1, IL6, IL-1β, and TNFα remained unaltered [36]. Conversely, brain analyses of rats selectively bred for their high ethanol consumption show increased Iba1 immunoreactivity in the brain reward system and increased levels of MCP1 in the hippocampus, after chronic voluntary consumption compared to water-exposed rats [37,38,39], possibly suggesting that high ethanol consumption is required to observe elevations in MCP1 levels.

Human studies have shown similar disparities. Postmortem brain analyses of AUD patients showed increased MCP1 mRNA levels in various areas of the brain reward system compared to controls [40,41,42]. However, a radioligand tracer associated with microglial activation and the occurrence of neuroinflammation has been shown to be reduced or unchanged in the brain of AUD patients that consumed ethanol chronically [43,44,45]. Nonetheless, this observed lack of changes may be attributed to the emergence of confounding targets of that particular ligand tracer ([11C]PBR28) after ethanol exposure [46]. These findings underscore the ongoing uncertainty regarding the significance of neuroinflammation in ethanol compulsive consumption.

In the present study, we aimed to assess changes in MCP1 levels in a model of intermittent voluntary chronic ethanol consumption in C57BL/6 mice, focusing on areas related to ethanol consumption motivation—the prefrontal cortex, hippocampus, and striatum—and the cerebellum, a brain region highly susceptible to ethanol exposure that deteriorates in chronic alcohol users [47]. Additionally, we also purified microglia from the hippocampus and striatum to uncover potential microglial-driven changes in MCP1 levels masked in homogenates. Our results show increased MCP1 mRNA levels in the cerebellum but not in the prefrontal cortex, hippocampus, or striatum of ethanol-consuming animals compared to controls, findings that were further supported by increased levels of another proinflammatory marker in the cerebellum, specifically the proinflammatory microRNA miR-155. These results suggest a higher sensitivity of the cerebellum to the proinflammatory effect of ethanol than other brain areas, potentially contributing to the severe cerebellar degeneration observed in some patients with AUD.

## 2. Results

### 2.1. Intermittent Chronic Voluntary Ethanol Consumption Model in C57BL/6 Female Mice

As an animal model of voluntary high ethanol consumption, female C57BL/6 mice were exposed every other day to a two-bottle choice between tap water or 15% ethanol dissolved in tap water for 67 days. Ethanol consumption escalated from the first day onwards, stabilizing around day 40 at over 15 g/kg/day, with over 80% ethanol preference in volume (Figure 1A–C). The average blood ethanol concentration the morning after the last ethanol consumption session was 26 mg/dL (Figure 1D). Notably, due to the natural fluctuations of voluntary consumption throughout the day despite consistent daily consumption, blood ethanol measurements were low or undetected in some mice, and a maximum exceeding 70 mg/dL was observed in one mouse (Figure 1D).

### 2.2. RT-qPCR MCP1 Analyses

To assess ethanol-induced neuroinflammation, RNA extracted from homogenates was used for RT-qPCR analyses, and the expression of MCP1 was selected as the main neuroinflammation marker, being the most consistent cytokine found to be increased in animal models of forced ethanol exposure [18,19,20,21,22,23,24,25,26,27,28,29]. The results show that MCP1 mRNA was significantly increased in the cerebellum of ethanol-exposed mice, while no significant changes were observed in prefrontal cortex, hippocampus, or striatum compared to water-exposed controls (Figure 2). 

### 2.3. Analyses of Isolated Microglia

Further investigation focused on whether MCP1 alterations occurred in microglia, the main MCP1-producing cells [17], to study whether significant differences in MCP1 might be masked by the contribution of other cells in the homogenates. However, the RT-qPCR analysis of isolated microglia from the hippocampus and striatum showed no alterations in MCP1 levels between experimental groups (Figure 3A). The non-responsiveness of microglia to ethanol was further evidenced by analyzing the mRNA levels of Iba1, the cytoskeleton-related protein associated with proinflammatory activation. Indeed, no significant changes were observed in microglia isolated either from the hippocampus or the striatum (Figure 3B). In addition, a flow cytometry analysis to determine whether the percentage of CD11b+, CD45^high^ cells, usually reported as activated microglia or infiltrating macrophages [48,49], showed no changes in the hippocampus or striatum of mice that consumed ethanol intermittently, chronically, and voluntarily compared to control animals (Figure 3C,D). Thus, cerebellum homogenates were the only brain region where neuroinflammation was observed in this study design.

### 2.4. Cerebellum RT-qPCR Characterization

To further characterize the degree of neuroinflammation observed in the cerebellum, the mRNA levels of other proinflammatory proteins and proinflammatory (miR-155 [50,51]) and anti-inflammatory (miR-21, miR-146a, and Let-7d [52,53]) microRNAs were measured. The mRNA levels of proinflammatory proteins interleukin 6 (IL6), cyclo-oxygenase 2 (COX2), and Toll-like receptor 4 (TLR4), and of chemokines MIP1α, MCP5, and CXCL2 and the chemokine receptor CCR2, were unchanged (Figure 4A). Interestingly, the levels of the proinflammatory miRNA miR-155 were increased in the cerebellum of mice that consumed ethanol compared to controls, while no changes in the selected anti-inflammatory microRNAs were observed (Figure 4B). An increase in miR-155 levels has been previously reported in the cerebellum of mice exposed to ethanol as the sole source of food compared to pair-fed mice [19].

## 3. Discussion

Our study aimed to uncover brain alterations associated with, and possibly instrumental to, the motivation for high ethanol consumption. We focused on MCP1, a chemokine that is a hallmark of brain neuroinflammation after forced ethanol consumption. Indeed, MCP1 is increased after ethanol exposure in studies that employ a nonelective ethanol bottle, repeated ethanol gavage, or repeated ethanol systemic administration [18,19,20,21,22,23,24,25,26,27,28,29]. Therefore, we analyzed the MCP1 mRNA levels in the brain of C57BL/6 mice that voluntarily and chronically consumed ethanol every other day, a model known for inducing high voluntary ethanol consumption and high blood ethanol concentration [54,55]. This model also induces a prefrontal cortex transcription profile similar to the proinflammatory response observed after LPS administration, albeit without classic proinflammatory cytokine alterations in that brain area [56]. In our hands, C57BL/6 mice reached an ethanol consumption of over 15 g/kg/day after two months of intermittent voluntary ethanol consumption, which is similar to what was observed by other research groups using the same or a similar model [36,54,55,57]. Our results showed a significant rise in MCP1 mRNA levels in the cerebellum. However, we could not detect alterations in MCP1 mRNA levels in brain areas associated with motivation for ethanol consumption.

To further study the hippocampus and striatum, we analyzed MCP1 levels in RNA samples obtained from purified microglia. We adopted this approach based on the study by McCarthy et al. [36] which showed that changes that occur in microglia may be masked in homogenates and could be revealed after specific microglia purification before RNA extraction. However, we did not see changes in the mRNA levels of MCP1 in microglia obtained from the hippocampus or the striatum of mice that voluntarily consumed ethanol compared to control animals. These results are similar to the findings of the study by McCarthy, which did not find an alteration of the levels of MCP1 in microglia obtained from the prefrontal cortex of mice that underwent the same ethanol consumption model used in this study [36].

Our study adds to a body of research failing to detect transcriptional differences in proinflammatory genes in animal models of AUD characterized by voluntary ethanol consumption. Studies of elective ethanol consumption in rats and mice have reported mild or absent proinflammatory changes in brain areas related to ethanol consumption motivation [34,35,36]. Conversely, studies using selectively bred rats for their high ethanol consumption have shown increased MCP1 levels in the hippocampus [37]. The reason for these disparities remains unclear, though the selection that induced high ethanol consumption in selectively bred rats may have generated a particular sensitivity to develop alterations that promote the motivation for drug consumption, which may include neuroinflammation, as it has been recently discussed [58].

Contrasting with voluntary ethanol consumption, neuroinflammatory alterations are prominently observed after forceful ethanol exposure, usually reporting an increase in classic proinflammatory cytokines in the brain of animal models of AUD [18,19,20,21,22,23,24,25,26,27,28,29]. A hypothesis for the cause of notable neuroinflammatory alterations in studies that employ forceful ethanol administration is the high blood ethanol concentration that can be reached after high amounts of ethanol are administered or consumed. In humans, high ethanol consumption or binge drinking is defined as consuming >4 drinks for females or >5 drinks for males within a short 2 h period, resulting in a blood ethanol concentration of over 80 mg/dL [59]. Studies reporting moderate-to-high blood ethanol concentration after ethanol exposure (>80 mg/dL on average) often show a significant increase in the levels of major proinflammatory markers, such as TNFα, IL6, or IL-1β [23,28,31,60]. Meanwhile, studies that report lower blood ethanol concentrations do not show changes in levels of classic proinflammatory molecules such as IL-1β and TNFα [27,30,36] but do show an increase in MCP1 levels in the prefrontal cortex [27]. Our findings align with this pattern, as we found a low blood ethanol concentration of 26 mg/dL the morning after the last ethanol voluntary consumption session. However, consistent neuroinflammation after high blood ethanol concentrations is not universally observed across different animal models of AUD. For example, intermittent vapor ethanol exposure, which sensitizes rats to voluntarily consume more ethanol, results in blood ethanol concentrations up to 250 mg/dL [45,61]. Nevertheless, neuroinflammation analyses using this model show only mild or no alterations in the levels of neuroinflammation markers in rats [45,61]. Similarly, even when using animal models of forced ethanol exposure significant neuroinflammation is not always observed. Marshall et al. employed rats given 5 g/kg ethanol daily by gavage, resulting in a blood ethanol concentration of over 300 mg/dL. This treatment showed increases in neuroinflammation markers such as Iba1 in the hippocampus, but this finding did not correlate with a rise in the levels of the complete proinflammatory profile, with cytokines IL6 and TNFα remaining unchanged [62]. In addition, studies in mice by Kane et al. employing 6 g/kg/day ethanol given by gavage for 10 days report blood ethanol concentrations exceeding 300 mg/dL [20,21]. Yet, only mild neuroinflammation was reported, showing increased mRNA levels of MCP1 in the hippocampus and cerebellum and increased mRNA levels of IL6 in the cerebellum, among adult and aged mice, while other markers of neuroinflammation, such as TNFα, were reported unaltered [20,21]. On the other hand, MCP1 and MCP1 receptor CCR2 knockout mice consume less ethanol voluntarily compared to their wild-type counterparts and show a higher aversion to ethanol in a conditioned place aversion test [63]. However, being a systemic knockout model, it is not clear whether the reduction in ethanol consumption is a result of reduced MCP1 in brain areas associated with ethanol consumption motivation. Overall, the onset of neuroinflammation seems to depend on the model used, and further investigation is required to unveil what causes the different outcomes observed. However, our results and those of others suggest that significant neuroinflammation may not be necessary to promote high ethanol consumption and preference in animal models of AUD. We acknowledge that although we did not find changes in the MCP1 mRNA levels in the brain reward system in our study, it is conceivable that such alterations might have occurred in a different stage in the development of ethanol consumption motivation, prior to the brain analysis at the 67-day mark and later normalized. However, existing research indicates that neuroinflammation once initiated can persist for months, as demonstrated using LPS administration [64]. Furthermore, while the protein levels of MCP1 were not analyzed in this study, other research indicates that ethanol-induced neuroinflammation typically involves alterations in MCP1 mRNA levels [18,19,20,21,23,26,28], which we did not observe. Additionally, it is worth considering the possibility that our observations could be sex-dependent, as ovulation and luteolysis promote the peripheral increase in MCP1 mRNA levels [65], which could have altered our results if the same phenomena occur in the brain. However, studies have shown ethanol-induced neuroinflammation in studies using male mice [18], female mice [19], and both [23] under forced ethanol consumption, suggesting that both sexes can undergo ethanol-induced neuroinflammation. Therefore, we can conclude that changes in the MCP1 mRNA levels in the brain reward system are not required for the motivation for ethanol voluntary consumption after two months of consumption in female mice.

Furthermore, in addition to the increase in cerebellar MCP1 mRNA levels, our investigation revealed increased levels of miR-155, a potent modulator of the immune response [50,51]. However, the levels of classic proinflammatory cytokines such as TNFα, IL-1β, and IL6 remained unaltered. Interestingly, the mentioned studies by Kane also report that the cerebellum is more sensitive to ethanol-induced neuroinflammation compared to the other brain areas studied [20,21]. This aligns to the findings in the present study, despite the completely different models employed by Kane and us. Moreover, the cerebellum has been the subject of several studies that sought to find changes in the levels of proinflammatory markers. It has been shown that the forced consumption of 5% ethanol in a liquid diet for 5 weeks [19] or repeated systemic administrations of ethanol for 25 days plus one instance of 5 g/kg ethanol gavage [25] increases the mRNA levels of MCP1, TNFα, IL-1β, and miR-155 in the cerebellum of mice compared to unexposed controls. Contrasted to the lack of alterations we found in other areas of the brain, the findings in the cerebellum suggest a special sensitivity of this area to ethanol-induced neuroinflammation. Indeed, in humans, the cerebellum is especially vulnerable to ethanol’s deleterious effects [47]. Acute ethanol can induce cerebellar ataxia, characterized by impaired posture, and ataxic gait and dysarthria (scanning speech), even at low blood ethanol concentrations [66]. On the other hand, chronic ethanol consumption induces cerebellar atrophy, characterized by the progressive loss of Purkinje cells [67]. This condition develops in 10 to 30% of AUD patients [68,69,70,71] and correlates more significantly to the chronicity of ethanol exposure rather than the amount consumed [70]. However, deleterious cerebellar effects have been suggested to be associated with the malnutrition of AUD patients [69]. Our findings in the present study and others [19,20,21,24,25] suggest that the particular cerebellar sensibility to ethanol-induced neuroinflammation may contribute to cerebellar impairment, which had also been proposed before as a mechanism for cerebellar damage [67]. Studies that investigate the consequences of forced ethanol consumption show significant neuroinflammation in the cerebellum, including alterations in microglia, astrocytes, and oligodendrocytes in mice [24], while some of those alterations are shown to be dependent on TLR4 signaling [19,25]. However, it is not clear whether the same pathways would be altered after voluntary consumption or whether this mechanism would reflect a distinct cerebellar sensitivity. In addition, while our study did not ascertain whether cerebellar alterations correlated with signs of cerebellar ataxia, our findings suggest that interventions aimed to reduce cerebellar neuroinflammation might be beneficial to prevent the onset of ethanol-induced cerebellar ataxia or atrophy. Moreover, several pharmacological agents have been shown to reduce cerebellar neuroinflammation and improve cerebellar function in animal models of cerebellar ataxia [72,73]. The efficacy of such interventions to treat ethanol-derived cerebellar neuroinflammation should be addressed in a future study. Overall, these findings imply that the consistent consumption of ethanol, even if it results in low blood ethanol concentration, may be harmful to the cerebellum. This is particularly concerning given the ongoing debate regarding the possible health benefits of low-to-moderate ethanol consumption [74].

## 4. Materials and Methods

### 4.1. Animals

Female C57BL/6J mice of 6 weeks of age were obtained from the Universidad del Desarrollo vivarium. Female mice were selected due to their higher voluntary ethanol consumption, more consistent consumption pattern, and the attainment of a higher blood ethanol concentration compared to male mice [55,75,76]. Mice were individually housed in a 12 h/12 h light/dark daily cycle (lights were on at 8 a.m.) with ad libitum access to water and food, in cages within close proximity to one another. All animal procedures adhered to the guidelines of the Universidad del Desarrollo Animal Care Committee (CICUAL, approval DCIM-2021/03).

### 4.2. Ethanol Voluntary Consumption Design

Mice were randomly assigned to either ethanol or control groups. Those in the ethanol group were exposed every other day to two graduated pipettes, one containing tap water and the other containing 15% ethanol (1009832500, Merck, Darmstadt, Germany) in tap water. Pipettes were provided at 2 pm and removed at the same time the day after, with the position of the ethanol and water pipettes alternated for each new consumption session to avoid a side preference. Control animals were solely exposed to the graduated water pipette. Sentinel pipettes for both water and ethanol solutions were installed in empty cages to monitor spontaneous volume reduction, resulting in a 0.2 to 0.4 mL loss. This volume was deducted from the consumption calculations in accord with the volume loss of each sentinel pipette each session. This protocol was maintained for 67 days. Next, immediately after the last ethanol consumption day, mice were removed from their cages at 8 am and blood from their tail vein was collected for ethanol blood concentration determination, after which the euthanasia procedure was performed. Blood samples were stored sealed at 4 °C, and the ethanol concentration was determined the following day, using an alcohol reagent kit (GMRD-113, Analox Instruments, Hammersmith, UK) and an AM1 Alcohol Analyzer (Analox).

### 4.3. Euthanasia

Animals were anesthetized using 4% sevoflurane in oxygen. Next, animals were transcardially perfused using ice-cold phosphate-buffered saline (PBS, 70011044, Thermo Fisher Scientific, Cleveland, OH, USA). Following perfusion, the brain was extracted, and the prefrontal cortex, hippocampus, striatum, and cerebellum were dissected. For microglia purification, the hippocampus and striatum from one hemisphere were kept in dissection media (see below), while the rest of the dissected tissues were snap frozen in liquid nitrogen.

### 4.4. Microglia Purification by FACS and RNA Extraction

Microglia were isolated using fluorescence-activated cell sorting (FACS), following the protocol described by Pan and Wan (2020) [77]. After dissection, the hippocampus or striatum from one hemisphere was minced and dissociated into a single-cell suspension using the neural tissue dissociation kit with papain (130-092-628, Miltenyi Biotec, Bergisch Gladbach, Germany). Cells were separated from myelin using Debris Removal Solution (130-109-398, Miltenyi Biotec) to prepare the cell suspension and were resuspended in PBS. Cells were incubated with anti-CD16/32 in PBS and 2% fetal bovine serum (FBS, SH30396.03, HyClone, Logan, UT, USA), followed by incubation with anti-CD11b and anti-CD45 antibodies, and then resuspended in Versene (0.02% ethylenediaminetetraacetic acid (EDTA) in PBS), 2% FBS, and 0.2% RNAsin (N2511, Promega, Madison, WI, USA). Cells were sorted by FACS using a FACSAriaIII cell sorter (BD Biosciences, San Jose, CA, USA). Microglia (CD11b+, CD45^low/mid^) were directly collected in Trizol LS (10296028, Thermo) and stored at −80 °C. Then, RNA from microglia stored in Trizol LS was extracted using the RNA Clean Up and Concentration Kit (23600, Norgen Biotek Corporation, Thorold, ON, Canada) following the manufacturer’s instructions, yielding a 20 μL sample and stored at −80 °C. All antibodies were from Biolegend (San Diego, CA, USA), other reagents were from Merck.

### 4.5. Tissue Homogenization and RNA Extraction

Frozen tissue from the prefrontal cortex, hippocampus, striatum, or cerebellum was thawed in Trizol (15596018, Thermo) and immediately homogenized using a pellet pestle homogenizer (Kimble). The RNA-containing aqueous layer was obtained by the addition of chloroform (Merck; 1/5 of Trizol initial volume) and 12,000× *g*, 15 min, 4 °C centrifugation. The aqueous layer was collected in new tubes and 20 μg of molecular-grade glycogen (R0561, Thermo) was added to each sample. Isopropanol (Merck, 1/2 of Trizol initial volume) was added to each sample and incubated overnight at −20 °C to promote RNA precipitation. Samples were centrifuged at 12,000× *g*, for 10 min at 4 °C, and an RNA pellet was obtained. Pellets were washed using 75% ethanol, resuspended in 20 μL RNAse/DNAse free water (10977-015, Invitrogen, Carlsbad, CA, USA), and stored at −80 °C.

### 4.6. cDNA Synthesis for mRNA Determination by RT-qPCR

For tissue homogenates, 1 μg of RNA measured by spectrophotometry using Nanodrop (Thermo) was used for cDNA synthesis using M-MLV Reverse Transcriptase (Invitrogen). For isolated microglia from the hippocampus and striatum, the RNA yield was low, and its concentration was undetected by spectrophotometry or by a High Sensitivity mRNA Assay Kit (Q32852, Invitrogen); therefore, 8 μL of RNA samples was used for cDNA synthesis using M-MLV Reverse Transcriptase (Invitrogen). mRNA amplification was determined by RT-qPCR using SYBR green (Brilliant II SYBR green master mix, Agilent) and normalized to the mRNA levels of glyceraldehyde-3-phosphate dehydrogenase (GAPDH). This strategy allowed for analyses of mRNA levels from purified microglia, and the threshold cycle of GAPDH was 26.7 ± 0.6 and 23.5 ± 0.8, for samples obtained from the hippocampus and striatum, respectively. Primer sequences are listed in Appendix A.

### 4.7. miQPCR Method for Determination of microRNA Levels by RT-qPCR

MicroRNA levels were assessed using RNA extracted from cerebellar homogenates following the miQPCR method described by Benes et al. (2015) [78]. This protocol involved the ligation of RNA molecules to a 26-base-long oligonucleotide modified at the 5′ end with a 5′,5′-adenyl group (sticky end) and at the 3′ with a dideoxycytidine group (blunt end), called miLinker and synthetized at IDT, allowing for the short microRNAs (~20 bases) to be elongated sufficiently for retrotranscription. The elongation was attained by incubating 10 ng of total microRNA, quantified by Qubit microRNA Assay Kit (Q32881, Invitrogen), with 15 μM miLinker, 200 U/μL T4 RNA Ligase (NEB), 5 mM MgCl2, 40 U/µL RNasin (Promega), 17.2% polyethyleneglycol, and 1X NEB buffer at 25 °C for 30 min. The first strand was synthetized by first adding the first-strand primer which binds to miLinker (miQRT, 10 μM) and dNTPs (10 mM), incubating at 85 °C for 3 min. Then, the SuperScript II retrotranscriptase (Invitrogen) and 10 mM dithiothreitol in 1X RT first-strand buffer (Invitrogen) were added and incubated at 46 °C for 30 min and 85 °C for 3 min. Samples were stored at −80°C until use. Subsequently, RT-qPCR was performed using SYRB green-based RT-qPCR determination, using a forward primer specific for each microRNA and a general reverse primer (Upm2a). MicroRNA levels were normalized to U6 small nuclear RNA (RNU6) levels. The primers and other sequences used are listed in Appendix A.

### 4.8. Statistical Analyses

Each analysis was conducted using samples obtained from seven mice per group, with the exception of samples lost from the same two control mice due to a technical error: purified microglia from the hippocampus and striatum were collected in the wrong solution after cell sorting, rending the number of samples for the control group of isolated microglia analysis to 5. The proportion of activated microglia (CD11b+, CD45^high^) obtained from the hippocampus and striatum relative to all microglia (CD11b+, CD45+) was quantified using FlowJo software version 8 (Tree Star, Inc., Ashland, OR, USA). RT-qPCR analyses were based on the 2^−ΔΔCt^ method. To evaluate the statistical significance of the differences between the comparisons of the control and the ethanol groups, Multiple Non-parametric Mann–Whitney tests were applied using the Holm–Šídák method to correct for multiple comparisons. A *p*-value < 0.05 denoted statistical significance. Outliers were removed when identified by the ROUT method [79], using Q = 0.5. GraphPad Prism v10 (GraphPad Software Inc.; San Diego, CA, USA) was used to perform all statistical analyses.

## 5. Conclusions

Is not clear whether the onset of neuroinflammation, and its hallmark marker MCP1, occurs in animal models of voluntary ethanol consumption, challenging the notion that neuroinflammation is pivotal in driving consumption motivation. Our findings suggest that brain areas involved in motivation for ethanol consumption, such as the prefrontal cortex, hippocampus, and striatum, appear resilient and do not show MCP1 alterations, which is also observed in isolated microglia, in C57BL/6 mice subjected to voluntary, intermittent, and chronic ethanol consumption. Conversely, we observed increases in MCP1 and miR-155 in the cerebellum, albeit without changes in classic proinflammatory markers, indicating mild neuroinflammation in this brain region. A higher cerebellar susceptibility to neuroinflammation may serve as a trigger for the cerebellar degeneration observed in late-stage alcoholism in humans.

## Figures and Tables

**Figure 1 ijms-25-04173-f001:**
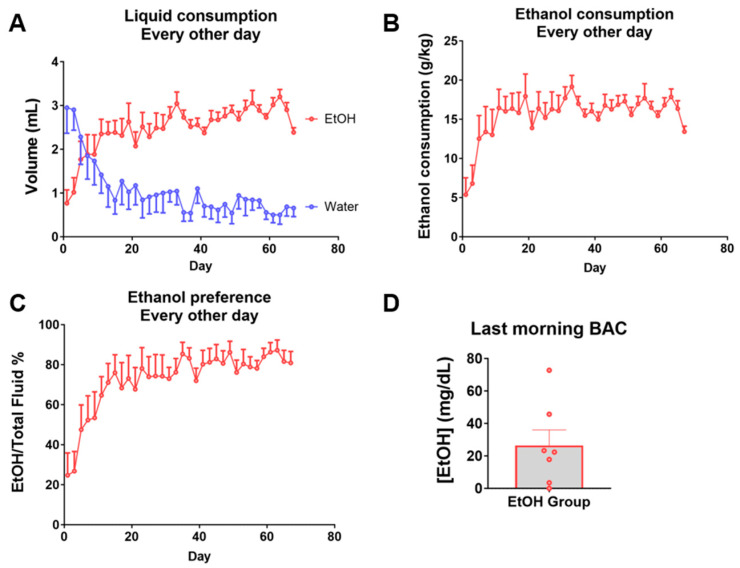
Mouse model of every-other-day voluntary ethanol consumption. (**A**) Every-other-day ethanol and water consumption of the ethanol group, in volume. (**B**) Every-other-day ethanol consumption normalized by body weight. (**C**) Every-other-day ethanol preference of the ethanol group as percentage of total fluid consumption. (**D**) Blood ethanol concentration from samples obtained in the morning after the last consumption session, before euthanasia. *n* = 7 per group. Data are presented as mean ± SEM.

**Figure 2 ijms-25-04173-f002:**
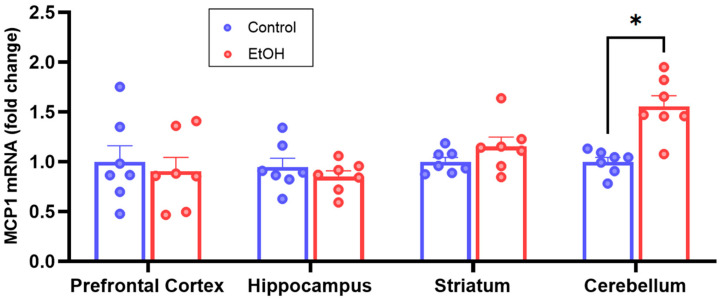
MCP1 mRNA levels in different brain areas of mice that consumed ethanol chronically and voluntarily compared to controls. MCP1 mRNA levels determined by RT-qPCR in homogenates of prefrontal cortex, hippocampus, striatum, and cerebellum obtained from mice that consumed ethanol chronically and voluntarily intermittently for 67 days and controls exposed to water only. MCP1 was normalized to GAPDH, using 2^−ΔΔCt^ method. *n* = 7 per group. Data are presented as mean ± SEM, * *p* < 0.05, Multiple Non-parametric Mann–Whitney tests.

**Figure 3 ijms-25-04173-f003:**
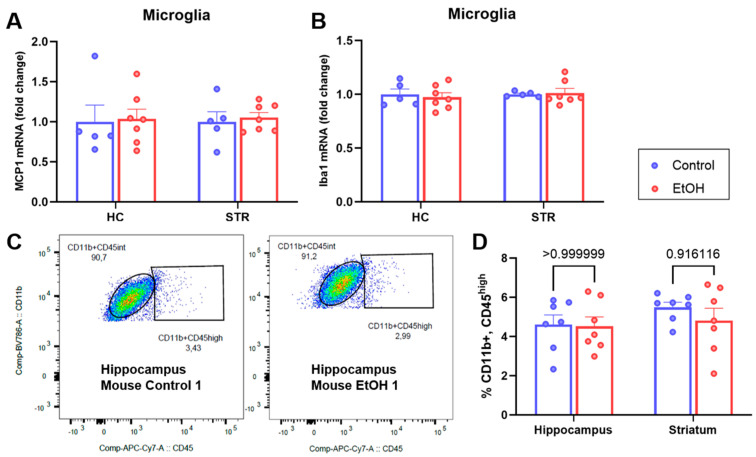
Analysis of proinflammatory markers in microglia isolated from hippocampus and striatum. MCP1 (**A**) and Iba1 (**B**) mRNA levels of microglia isolated from hippocampus (HC) or striatum (STR) of mice that consumed ethanol chronically, voluntarily, and intermittently compared to controls, determined by RT-qPCR. Gene expression was normalized to GAPDH, using 2^−ΔΔCt^ method. (**C**) Representative diagrams of the gating strategy showing the proportion of activated microglia (CD11b+, CD45^high^) as percentage of total microglia cells (CD45+, CD11b+) obtained from hippocampus and striatum (**D**). *n* = 5−7 per group. Data are presented as mean ± SEM. Multiple Non-parametric Mann–Whitney tests.

**Figure 4 ijms-25-04173-f004:**
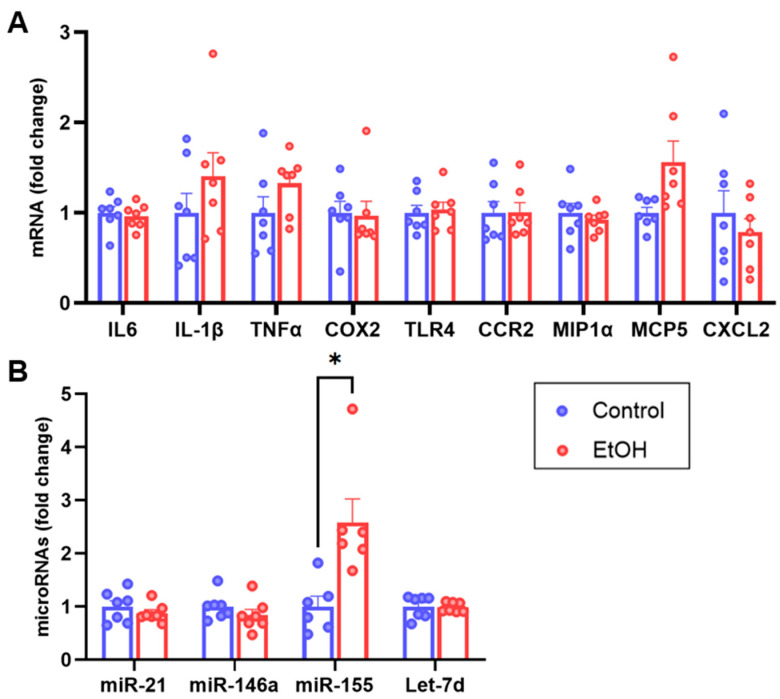
Inflammatory characterization of cerebellum homogenates from mice that consumed ethanol chronically and voluntarily. mRNA levels determined by RT-qPCR of selected classic proinflammatory genes IL6, IL-1β, TNFα, COX2, and TLR4; chemokines CCR2, MIP1α, MCP5, and CXCL2 (**A**) and proinflammatory (miR-155) and anti-inflammatory (miR-21, miR-146a, and Let-7d) microRNAs (**B**). Gene expression was normalized to GAPDH for mRNAs and to RNU6 for microRNAs, using 2^−ΔΔCt^ method. *n* = 7 per group. Data are presented as mean ± SEM, * *p* < 0.05, Multiple Non-parametric Mann–Whitney tests.

## Data Availability

All data supporting this study are included within this article.

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
