# Peer review of "Two-Month Voluntary Ethanol Consumption Promotes Mild Neuroinflammation in the Cerebellum but Not in the Prefrontal Cortex, Hippocampus, or Striatum of Mice"

_ijms, 2024, doi:10.3390/ijms25084173_

Round 1

Reviewer 1 Report

Comments and Suggestions for Authors

Please see the attached PDF file.

Reviewer 2 Report

Comments and Suggestions for Authors

In this well-written study, the authors investigated a main marker of neuroinflammation, the microglia-derived monocyte chemoattractant protein 1 (MCP1) in animal models of alcohol use disorder in which ethanol is forcefully given. The authors studied MCP1 mRNA levels in areas implicated in consumption motivation, the prefrontal cortex, hippocampus, and striatum- as well as in the cerebellum, a brain area highly sensitive to ethanol, from C57BL/6 female mice subjected to intermittent and voluntary ethanol consumption for two months. The authors found a significant increase in MCP1 mRNA levels in the cerebellum of mice that consumed ethanol compared to controls, whereas no significant changes were observed in the prefrontal cortex, hippocampus, or striatum; or in microglia isolated from the hippocampus and striatum. The authors measured also the expression changes in other proinflammatory markers and chemokines, revealing significant increases in MCP5 and the proinflammatory microRNA miR-155. The authors stated that the onset of marked neuroinflammation in motivation-related areas is not required for high voluntary consumption in C57BL/6 mice. The authors also speculated that cerebellar susceptibility to neuroinflammation may be a trigger to the cerebellar degeneration that occurs after chronic ethanol consumption in humans.

The paper is crystal clear, the data presentation is solid and in general methods and results are well described.

The main limitation of this paper is the use of C57BL/6 female mice.

I’m wondering why the authors used female mice to analyze MCP1 since a plethora of previous studies clearly showed CCL2 modifications during ovulation/luteolysis.

Did the authors consider the estrous effect of the animals in the absence of pregnancy?

Comments on the Quality of English Language

Minor editing of the English language required

Reviewer 3 Report

Comments and Suggestions for Authors

Review of "Two-Months Voluntary Ethanol Consumption Promotes Mild Neuroinflammation in the Cerebellum, but Not in the Prefrontal Cortex, Hippocampus, or Striatum of Mice"

Introduction.

The manuscript addresses a significant issue concerning the relationship between chronic ethanol consumption and neuroinflammation, particularly focusing on the role of MCP1 in voluntary ethanol consumption. The introduction provides a comprehensive overview of alcohol use disorder (AUD), emphasizing the importance of understanding neuroinflammatory processes in promoting ethanol consumption. The background information sets a strong foundation for the study's objectives.

Methods.

The methods section describes the experimental design in detail, including the animal model used, ethanol exposure regimen, and analytical techniques. The use of female C57BL/6 mice and intermittent voluntary ethanol consumption reflects a relevant model for studying human AUD. However, the high variability in ethanol consumption observed among mice may present challenges in data interpretation and reproducibility. 

Results and Discussion.

The results section presents clear findings regarding MCP1 mRNA levels in different brain regions following ethanol consumption. The significant increase in MCP1 expression in the cerebellum suggests a specific susceptibility to ethanol-induced neuroinflammation in this region. The inclusion of other proinflammatory markers and microRNAs provides a comprehensive understanding of neuroinflammatory changes associated with ethanol exposure. The authors effectively interprets the results in the context of existing literature highlighting the significance of the findings, as well as acknowledge the limitations of the study, such as the lack of transcriptional differences in other brain regions and the variability in ethanol consumption among mice. The comparison with previous studies strengthens the discussion and underscores the complexity of neuroinflammatory responses to ethanol across different models.

Conclusion.

The conclusion provides a concise summary of the key findings and their implications for understanding ethanol-induced neuroinflammation. The discussion on the cerebellum's susceptibility to neuroinflammation adds novel insights to the field and raises important questions about the long-term consequences of chronic ethanol consumption.

Suggestions for Improvement.

1.     Statistical Analysis: Provide additional information on statistical methods employed, including the significance thresholds and corrections for multiple comparisons, to enhance the rigor of data interpretation.

2.     2. Reproducibility: Discuss potential sources of variability in ethanol consumption among mice and strategies to mitigate these variations to improve reproducibility across experiments.

3.     Mechanistic Insights: Consider exploring potential mechanisms underlying the cerebellum's sensitivity to ethanol-induced neuroinflammation, such as the involvement of specific cell types or signaling pathways, to deepen understanding of the observed effects.

4.     Clinical Relevance: Discuss the translational implications of the findings for understanding cerebellar dysfunction in AUD patients and potential therapeutic targets for mitigating neuroinflammatory responses.

5.     Future Directions: Propose future research directions, such as investigating the temporal dynamics of neuroinflammatory changes or evaluating the efficacy of anti-inflammatory interventions, to address remaining questions and advance the field.

Overall, the manuscript provides reasonably good insight into the relationship between chronic ethanol consumption and neuroinflammation, with particular emphasis on the cerebellum. Addressing the suggested improvements would further increase the sustainability and impact of the study.

Round 2

Reviewer 1 Report

Comments and Suggestions for Authors

I have no further comments to the authors.  

Author Response

Thank you for contributing to making this a much better manuscript.

Reviewer 2 Report

Comments and Suggestions for Authors

The authors made great efforts to improve the paper. However, since ovulation or luteolysis may alter peripheral MCP1 levels, it cannot be excluded that in female mice these factors could modify also brain MCP1 mRNA levels. I do believe that this issue is a limitation of the study and should be included in the paper. 

Author Response

Thank you for your kind words. The limitation was detailed as suggested. An explanation was added to Page 8, Lines 273 to 276:
"Additionally, it is worth considering the possibility that our observations could be sex-dependent, as ovulation and luteolysis promote the peripheral increase of MCP1 mRNA levels (Nio-Kobayashi, Junko, et al. 2015), which could have altered our results if the same phenomena occur in the brain."

The references list was updated accordingly.

Round 3

Reviewer 2 Report

Comments and Suggestions for Authors

The paper is ok right now.